environmental chemistry/analytical chemistry

electrochemistry, sensor, electrode

**Authors for correspondence:**
Guochen Zhao
e-mail: zhaogch@sdas.org
Yuanfeng Huang
e-mail: huangyf@sdtj.sd.cn

# Monitor application of multi-electrochemical sensor in extracting bromine from seawater

Qiujin Wang[1], Jianbo Wu[1], Guochen Zhao[2], Yuanfeng Huang[3], Zhen Wang[3], Hao Zheng[1], Yifan Zhou[1], Ying Ye[1] and Reza Ghomashchi[4]

[1]Ocean College, Zhejiang University, Zhoushan 316000, People's Republic of China
[2]Shandong Provincial Key Laboratory of High Strength Lightweight Metallic Materials, Advanced Materials Institute, Qilu University of Technology (Shandong Academy of Sciences), Jinan 250000, People's Republic of China
[3]Shandong Special Equipment Inspection and Testing Science and Technology Co., Ltd, Jinan 250000, People's Republic of China
[4]School of Mechanical Engineering, The University of Adelaide, Adelaide, South Australia 5005, Australia

 QW, 0000-0001-9961-3533; GZ, 0000-0002-2493-6343; YH, 0000-0001-6446-2923

In this paper, a set of online measurement devices of multi-electrochemical sensor was investigated. Combined with industrial distributed control system, it was first applied in extracting bromine from seawater to realize the real-time adjustment of production process parameters. In the process of extracting bromine from seawater, the pH value of acidified raw brine, the addition amount of $Cl_2$ in the oxidation stage and the addition amount of $SO_2$ in the absorption stage are key parameters to control the whole production process. The multi-electrochemical sensor realized a rapid and high-throughput detection of the above parameters by integrating an all-solid-stage bromide ion selective electrode (Br-ISE), Eh electrode and pH electrode. The Br-ISE and the pH electrode were self-developed electrodes and the Pt electrode was Eh electrode. The pH electrode was used to control the addition amount of $H_2SO_4$ during the acidification of the brine. The Eh electrode was used to control the addition amount of $Cl_2$ during the oxidation stage and the addition amount of $SO_2$ during the absorption stage. The Br-ISE was used to monitor the $Br^-$ concentration change in the raw brine. Results showed the optimum range of Eh in the oxidation stage and absorption stage of brine were 950–1000 mV and 580–610 mV, respectively. The application of multi-electrochemical sensor in industrial bromine production can realize real-time control of material addition and save the cost of production.

# 1. Introduction

Bromine is an important industrial chemical, widely applied to flame retardants, fire extinguishing agents, photographic materials, medicines, pesticides, etc. [1]. There are many well-developed methods to extract bromine from seawater [2], such as the separation membrane technology [3,4], the ion exchange resin method [5], the steam distillation method [6] and the air-blowing method [7]. The air-blowing method can be divided into the alkali liquid absorption method and the acid liquid absorption method depending on the different absorbents employed [8–10]. In China, the air-blowing method with acid liquid absorption occupies the largest market in extracting bromine from seawater, adopted by 90% of bromine factories [11]. The main process can be summarized as five steps (figure 1). Step 1, acidification: $H_2SO_4$ is added to the concentrated seawater to stabilize the pH of the concentrated seawater at 3–3.5. Step 2, oxidation: the $Br^-$ ions are oxidized to $Br_2$ by blowing $Cl_2$ into the acidified seawater. Step 3: air is blown out, the oxidized seawater is piped to the air-blowing tower and the free $Br_2$ is then blown out. Step 4, absorption: the air containing a large amount of free $Br_2$ is introduced into the absorption tower and $SO_2$ is injected. The free $Br_2$ in the air is transformed into hydrobromic acid $(SO_2 + Br_2 + 2H_2O = 2HBr + H_2SO_4)$. Step 5, distillation: transferring hydrobromic acid into the distillation column, and then HBr is oxidized to $Br_2$ by pumping in $Cl_2$ from the bottom of the column $(2HBr + Cl_2 = 2HCl + Br_2)$ [12]. Eventually, bromine vapour is collected from the top of the column. After condensation and bromine–water separation, high-quality product bromine is obtained. In the whole process of extracting bromine from seawater, the addition amount of $H_2SO_4$ in the acidification of brine, the addition amount of $Cl_2$ during the oxidation of brine and the addition amount of $SO_2$ during the absorption stage are the key parameters to determine the productivity of bromine. In Shandong Province of China, the number of bromide enterprises has exceeded 140. Among them, most manufacturers kept the traditional bromine production lines, mostly built in the 1970s and 1980s, until now [13], and contributed to dragged control system. The addition amounts of $H_2SO_4$, $Cl_2$ and $SO_2$ were controlled by manual titration every 2 h and the operational parameters were adjusted manually according to manual titration calculation results. Therefore, the results may fluctuate greatly within 2 h, resulting in a serious lag of regulation.

At present, there are few research studies working on the 'automation transformation' of bromine enterprises in China. According to the open literature, only a few manufacturers and research teams have made some useful explorations. In the oxidation stage, the problem to control the amount of chlorine added was studied by the China University of Petroleum and Shangdong Haihua Group (salt manufacturing factories). An online monitoring device based on redox potential was developed for the oxidation process, which realized the automatic control of chlorine gas [14]. Li *et al.* [15] adopted industrial ethernet and field bus technology and other network systems to build an automated monitoring system platform. Zhang *et al.* [16] designed a comprehensive automatic monitoring system consisting of Siemens S7300 PLC, S7-200 PLC controller, onsite instrument device and other hardware. Based on the potential analysis method of ultrasonic cleaning technology, Tang proposed online monitoring of the brine oxidation process to control the addition amount of $Cl_2$ and acid addition during the bromine production process. In our group, we independently developed a multi-electrochemical sensor integrated with bromide ion selective electrode (Br-ISE), pH electrode and Eh electrode, which is expected to automatically control the addition of $Cl_2$, $H_2SO_4$ and $SO_2$. Compared with the traditional liquid junction ISE, all-solid-state ion selective electrode shows obvious advantages including small size, anti-interference of external environment and easy to be integrated. All-solid-state ISE has been demonstrated as a promising direction of ISE research [17,18]. As early as 1980, ISEs were widely used as sensors in industrial manufacture for measuring the concentration of various ions, including ammonia, sodium, nitrate, fluoride, cyanide and sulfide [19–21], metal ion [22–28] and organics [29–33], which provided the basis for precise consistency control. Light [34] reviewed the industrial ISE systems that pH, calcium ion and fluorinion electrodes as real-time monitors had been used in actual industrial applications.

For all-solid-state ISEs, materials, such as glassy carbon [35], PVC filmogen [28,30,36,37], graphite [38] and silver strips [23], etc., were used as solid substrates. Based on our previous work, we fabricated a new all-solid-state Br-ISE-coated bromide ion-doped PANI electropolymerized film [39]. In addition, our research group successfully prepared an all-solid-state pH electrode-coated Ir/IrOH film on Ir wire [40]. We integrated the self-developed multi-electrochemical sensor with all-solid-state Br-ISE, pH electrode and Eh electrode to control the amount of $H_2SO_4$, $Cl_2$ and $SO_2$ during the extraction of bromine from seawater.

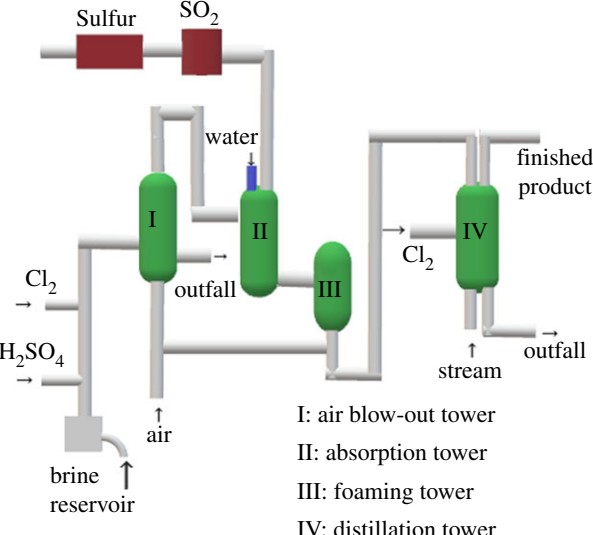

**Figure 1.** Technical process diagram of air-blowing bromine.

# 2. Critical control points of material delivery

## 2.1. The pH range of brine during the oxidation stage

The brine is alkaline (pH = 8) and $Br_2$ and $Cl_2$ can be hydrolysed easily in brine under such a condition. Hydrolysis will cause the loss of bromine in the form of HBrO and other oxides

$$Br_2 + H_2O = H^+ + Br^- + HBrO, \tag{2.1}$$
$$3Br_2 + 3H_2O = 6H^+ + 5Br^- + BrO_3^- \tag{2.2}$$

and
$$Cl_2 + H_2O = H^+ + Cl^- + HClO. \tag{2.3}$$

During the brine oxidation stage, the oxidation degree of the brine determines the final bromine extraction rate. If the concentration of $H^+$ in seawater is improved, then hydrolysis of bromine can be inhibited, it is also the purpose of acidification. When the pH of the brine is too low, a large amount of acid will be consumed, but the oxidation rate does not improve greatly, which will increase the cost. So, it is necessary to control the pH value of the oxidation liquid accurately. According to experiments reported in the open literature, the pH of the oxidized brine is usually stabilized at 3.5–4 [41].

## 2.2. The addition amount of $Cl_2$ during the brine oxidation stage

In the oxidation stage, after the brine is acidified with HClO, the $Br^-$ ion in the brine is replaced with a $Br_2$ by adding $Cl_2$. The reaction proceeds very quickly in the liquid phase, and the equilibrium is established instantaneously

$$Cl_2 + H_2O \rightleftharpoons HCl + HClO \tag{2.4}$$
$$HCl + HClO + 2NaBr \rightarrow 2NaCl + Br_2 \uparrow + H_2O. \tag{2.5}$$

In addition to the above main reactions, a series of side reactions will occur, such as the hydrolysis reaction of $Br_2$, $Cl_2$, $BrCl_2$ and BrCl carried out and the formation of complexes. When the supply of $Cl_2$ increases, the generated $Br_2$ will continue to react with $Cl^-$ ion to form a large amount of BrCl or polychlorinated bromine. Since BrCl and polychlorinated bromine are less volatile than $Br_2$, this will result in a lower blowing rate. If the amount of $Cl_2$ added is insufficient in the oxidation process of the raw material brine, the $Br^-$ ion cannot be oxidized to $Br_2$ and the oxidation rate is low. In theory, it takes 0.433 g of $Cl_2$ to oxidize 1 kg $Br_2$. The percentage of the ratio of the actual amount of chlorine to the theoretical amount of chlorine is generally referred to as the ratio of chlorine. Chlorine adding ratio is usually maintained at 105–115% in the air-blowing method. However, in order to ensure that the raw brine is fully oxidized, the workers usually put in too much $Cl_2$, which increases the cost of production [24].

## 2.3. The addition amount of $SO_2$ during the absorption stage

During the absorption process, the $SO_2$ is produced by sulfur combustion. In this process, the $Br_2$, $SO_2$ and water vapour in the absorption tower are thoroughly mixed and $SO_2$ is used as a reducing agent to reduce the $Br_2$ to HBr. The chemical reactions occur as follows:

$$SO_2 + H_2O = H_2SO_3 \tag{2.6}$$

$$Br_2 + H_2SO_3 + H_2O = 2HBr + H_2SO_4. \tag{2.7}$$

If the amount of $SO_2$ added is too small, the absorption of $Br_2$ is not achieved fully and therefore cannot be completely reduced. If the added $SO_2$ is more than the required amount, $SO_2$ cannot be fully used, which will lead to high cost. Therefore, controlling the timely addition of $SO_2$ needed to be resolved.

# 3. Experimental procedure

## 3.1. Reagents and apparatus

Ag wire (99.99% 0.5 mm in diameter) was purchased from the Precious Materials Company of Changzhou China. Sodium bromide, aniline, hydrochloric acid, sodium chloride, sodium sulfite, sodium sulfate, sodium nitrate, acetone, nitric acid and silver nitrate were obtained from Aladdin (Shanghai, China). All of the above chemicals were analytically of pure grade. The water used to configure the solution was ultrapure. The fabrication and electrochemical analysis were carried on an electrochemical workstation (IviumStat Dutch Ivium Technology BV Company). Three-electrode system was applied in the process: the auxiliary electrode was a $10 \times 10 \times 0.1$ mm Pt (99.99%) electrode (Chenhua, Shanghai, China) and the reference electrode was an Ag/AgCl reference electrode (CHI111, Chenhua, Shanghai, China). A FLUKE 123B industrial scopemeter and an SG102A function signal generator were used to generate half sine wave voltage. CNC ultrasonic cleaner KQ218 (Kunshan Ultrasonic Instrument Co. Ltd) was used to clean Ag wires and accelerate dissolution. Scanning electron microscopy (SEM SU-8010, Hitachi, Japan) was used to observe the surface topography of the coated Ag electrodes.

## 3.2. Preparation of electrodes

### 3.2.1. Preparation of pH electrode

Based on the previous study [39], the all-solid-state pH electrode was fabricated successfully. Using three-electrode system, Ir wire as the working electrode, the Ag/AgCl electrode as the reference electrode and the Pt as the auxiliary electrode, cyclic voltammetry of all three electrodes were scanned for three cycles at the scan rate of 50 mV s$^{-1}$ in an aqueous solution of 5% LiOH at the potential range of 0–0.9 V and the Ir(OH)x film was electrodeposited on the surface of the Ir wire. The electrode was cleaned in the deionized water and ethanol successively and naturally dried in air. Finally, the electrode was immersed in 3.5% NaCl solution for at least 4 h for improving the activation. In figure 2, the pH electrode has an excellent Nernst linear response (slope: $-55.5$ mV decade$^{-1}$, $R^2 > 0.9502$).

### 3.2.2. Preparation of Br-ISE

The Br-ISE coated by the bromine ion-doped PANI film was fabricated successfully using the cyclic voltammetry. The silver wire with nano-silver (30–60 nm) layer as the working electrode, the Ag/AgCl electrode as the reference electrode and the Pt as the auxiliary electrode. Cyclic voltammetry of all three electrodes were scanned for 10 cycles at the scan rate of 100 mV s$^{-1}$ in an aqueous solution of 1.0 M HCl and 0.3 M aniline at the potential range of $-0.2$ V to 0.9 V and then the three electrodes into a 0.1 M KBr solution and scanning for another 20 cycles at the potential range of $-0.2$ to 0.72 V. The prepared Br-ISE exhibited a wide linear dynamic range between $1.0 \times 10^{-1}$ and $1.0 \times 10^{-7}$ M with a near-Nernst slope of 57.33 mV decade$^{-1}$ (figure 3). In addition, we determined the response time (less than 1 s) impedance (300 Ω) and lifespan (over three months) of the prepared Br-ISE (based on the previous test).

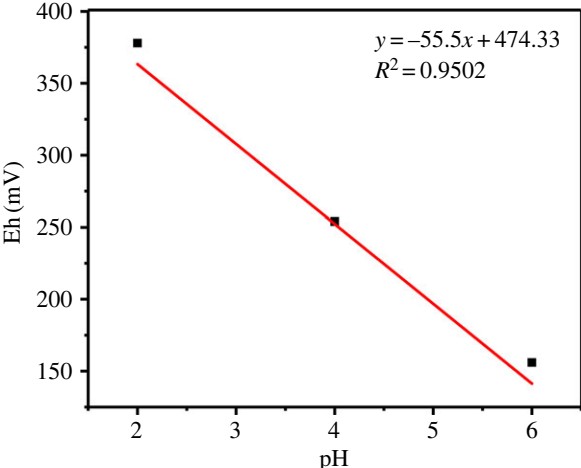

**Figure 2.** The calibration process and calibration curves of the pH electrode.

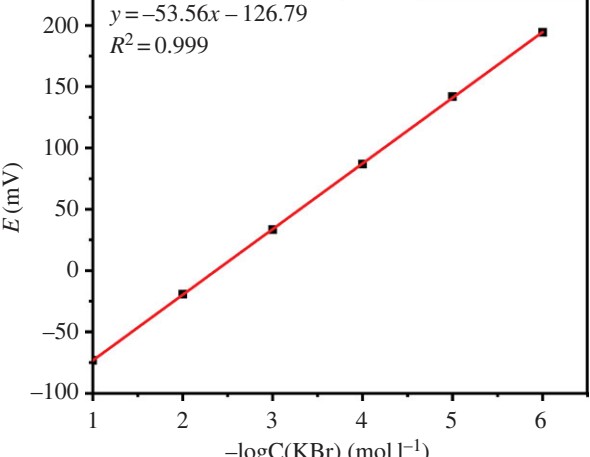

**Figure 3.** The calibration process and calibration curves of the Br-ISE.

### 3.2.3. Preparation of Eh electrode

In this study, the Pt wire was used to detect the Eh value [42].

## 3.3. Assembly of the multi-electrochemical sensor

In figure 4, all the prepared electrodes were assembled as a multi-electrochemical sensor. The multi-electrochemical sensor integrates a reference electrode, an Eh electrode and a pH electrode for each probe. The multi-electrochemical sensor consists of six parts: ISE probes, power interface, data transmission, datalog, power-reduced voltage module (from 220 V reduced to 24 V) and waterproof case.

# 4. Optimum parameters

## 4.1. The optimum range of pH during brine oxidation process

The bromine is extracted by air blowing in low concentration brine and the acidification pH value of the brine is controlled to about 3.5 [43]. When the pH value is higher than 4, the hydrolysis of bromine is intensified which affects the extraction rate. When the acidification pH value is less than 3, the consumption of $H_2SO_4$ is increased by more than 50%, but the oxidation rate is not improved.

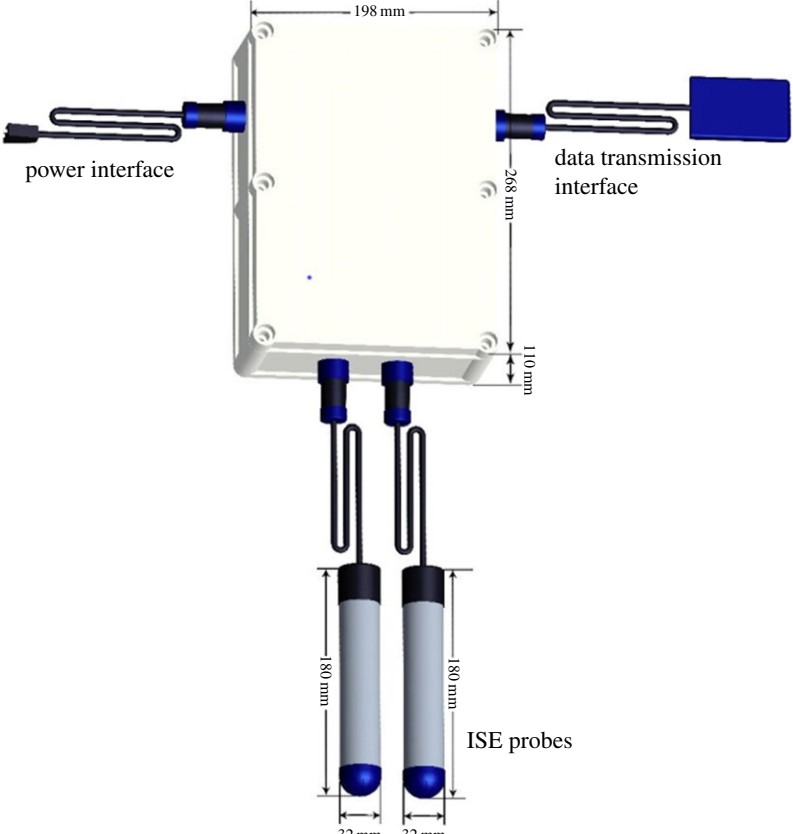

**Figure 4.** The sketch of the multi-electrochemical sensor.

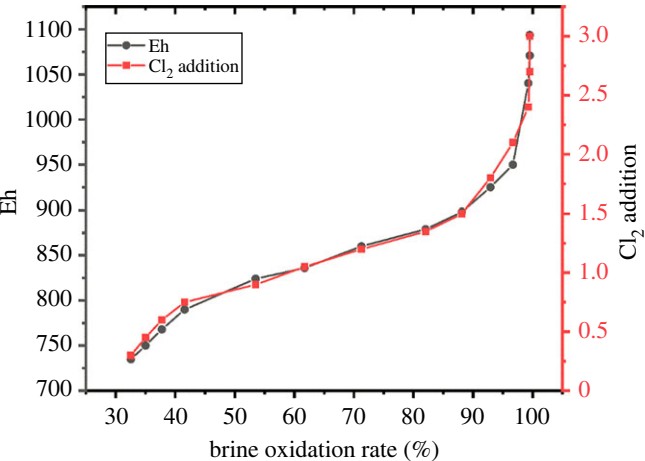

**Figure 5.** The sketch of the multi-electrochemical sensor.

## 4.2. The optimum range of Eh during brine oxidation process

In the laboratory environment, we simulated the chemical reaction process in the oxidation phase of the brine to obtain the optimum amount of $Cl_2$ added. Configuring a simulated brine solution with a bromide ion concentration of 300 mg l$^{-1}$ (2.52 mmol l$^{-1}$) the amount of $Cl_2$ added depends on the addition of HClO and HCl. Because HClO is a strong oxidant, the residual HClO concentration in the solution can be detected by the Eh electrode. Firstly, we adjust the pH of the simulated brine (50 ml) to the optimum value (pH = 3.5) and 0.1 M HClO and HCl were added drop-by-drop with 0.5 ml every time. After fully reacting the Eh and residual Br$^-$ ion concentration in the solution were tested by the Eh electrode and Br-ISE. In figure 5, with the addition of $Cl_2$, the oxidation rate of brine increases and when the amount of $Cl_2$ added is about 2.5, the oxidation rate reaches the highest value (oxidation rate = residual Br$^-$ content/300 mg l$^{-1}$).

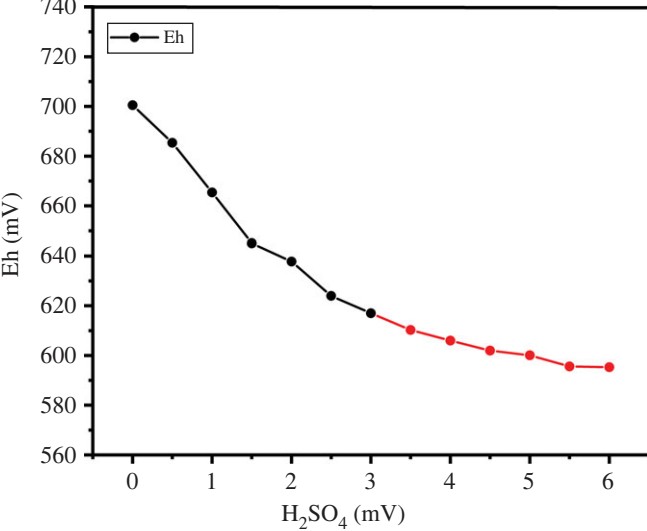

**Figure 6.** Relationship between Eh, Cl$_2$ addition and oxidation rate.

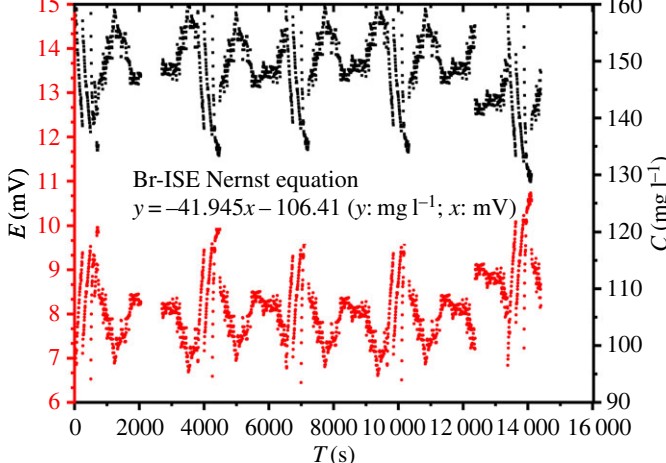

**Figure 7.** Simulation experiment in the absorption stage.

The value of oxidation rate of brine increased with the addition of Cl$_2$ and reached the highest value when the amount of addition of Cl$_2$ was at 2.5. Meanwhile, the value of Eh increased from 734 to 1000 mV; this shows that the Eh electrode can be used to guide the addition of Cl$_2$ in the brine oxidation stage. After repeated experiments, the brine oxidation rate reaches to its optimum value when the Eh value is controlled at 900–1000 mV.

## 4.3. The optimum range of Eh during the absorption process

Equation (2.7) is a redox reaction. The Br$_2$ was reduced to Br$^-$ by H$_2$SO$_3$ and the progress of the reduction of Br$_2$ to Br$^-$ can be judged according to the process of oxidation of H$_2$SO$_3$ to H$_2$SO$_4$. Firstly, 0.1 M concentration of H$_2$SO$_4$ (0.5 ml each time) was added to 50 ml of a 0.1 M H$_2$SO$_3$ solution and the change of the Eh value was observed simultaneously. In figure 6, the initial Eh value was 695 mV and gradually decreased to 580 mV with the addition of sulfurous acid and it became stable. According to this experiment, it is recommended to control the Eh value of the solution in the absorption tower at 580–610 mV.

# 5. Application

The multi-electrochemical sensor was applied initially at the bromine factory. The Haihua Group bromine factory is located in the northwest of Weifang City and is adjacent to Laizhou Bay in the Bohai Sea. This bromine plant has abundant underground brine resources and is the largest bromine production enterprise in China at present. The bromine plant uses the traditional experimental

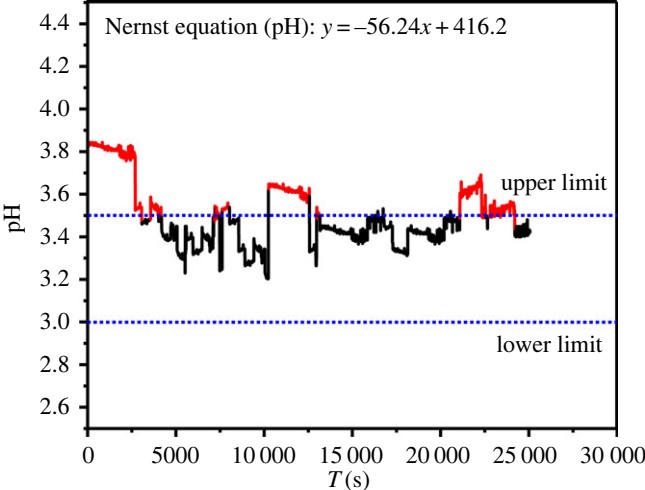

**Figure 8.** Data collected by the Br-ISE sensor within 7 h in the raw brine.

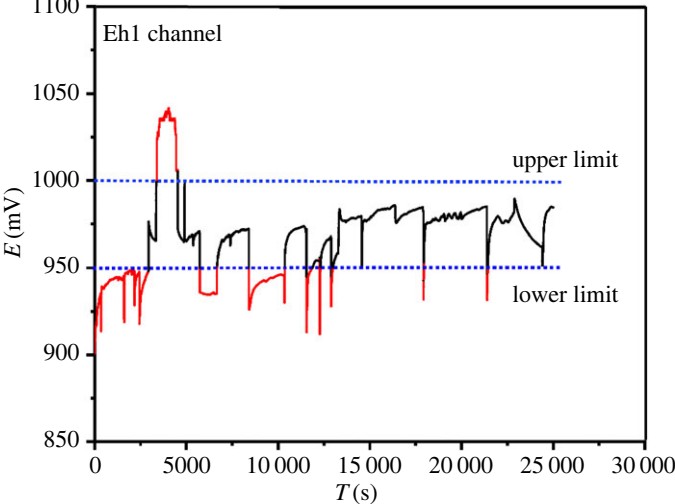

**Figure 9.** Data collected by the pH sensor within 7 h in the oxidation tower.

manual titration method, which leads to the adjustment lag in actual production. Therefore, in order to improve production efficiency and ensure production safety, it is necessary to upgrade the traditional onsite control to remote control and upgrade the timing sampling detection to real-time online monitoring. The amount of material added in each part of the production process is transformed into the voltage signal through sensor detection and is transmitted to the central control room through an industrial distributed control system (DCS) for displaying in real time.

The change range of potential and the corresponding bromine ion concentration were detected by the Br-ISE within 7 h real time in figure 7. The change range of the potential is 9.21–6.45 mV and the corresponding concentration change range was 162.88–139.98 mg l$^{-1}$. It can be seen from the change range of the potential and concentration that small potential drift has a great influence on the Br$^-$ ion concentration. It is notable that fluctuations in water flow friction of solid impurities in water and sloshing of electrodes can all affect the potential. The bromine concentration of the brine reservoir is about 147 mg l$^{-1}$ according to the practical calculation by manual titration. The maximum difference between the measured concentration of the sensor and the actual difference is 15.88 mg l$^{-1}$.

The pH electrode is relatively stable compared with Br-ISE and the Nernst slope is −56.24 mV decade$^{-1}$ with higher accuracy. In figure 8, the pH value of brine in oxidation tower is monitored by pH electrode within 7 h. The optimum oxidation pH range of brine is 3–3.5. If the amount of $H_2SO_4$ is too little, the indication of the pH electrode will immediately show a decrease (the red part in the figure 8); at the same time, the addition of sulfuric acid to the port electric valve will immediately increase the amount of sulfuric acid added to increase the pH of the oxidized brine.

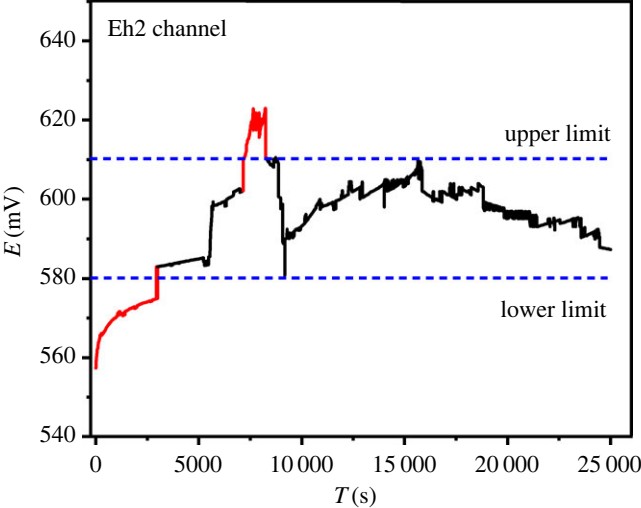

**Figure 10.** Data collected by the Eh sensor within 7 h in the oxidation tower.

The Eh of brine in oxidation tower within 7 h is given in figure 9. The optimum oxidation Eh range of brine is 950–1000 mV. When the value of Eh is below the lower limit of 950 mV or higher than the upper limit of 1000 mV, the chlorine electric control valve will automatically increase or decrease the $Cl_2$.

The Eh of brine in absorption tower within 7 h illustrated is in figure 10. The optimum Eh range of brine is 580–610 mV. When the value of Eh is below the lower limit of 580 mV or higher than the upper limit of 610 mV, the $SO_2$ control valve will automatically increase or decrease its concentration.

## 6. Conclusion

In this paper, we have tested the feasibility of applying the multi-electrochemical sensor control to industrial bromine production. The application of multi-electrochemical sensor integrated pH electrode, Eh electrode and Br-ISE in place of traditional manual titration has shown the ability of precise control of the $H_2SO_4$ feed inlet, the $Cl_2$ feed inlet and the $SO_2$ inlet. The optimum ranges of Eh in the oxidation stage and absorption stage of brine were 950–1000 mV and 580–610 mV, respectively, with the optimum range of pH in the brine acidification of 3–3.5. The input of these optimal control ranges into the DCS, as important control indicators of the system, can automatically adjust the additions according to the optimal range. With the advantages of *in situ* monitoring in real time, our multi-electrochemical sensor has the potential to improve the economics of the industrial bromine production.

Data accessibility. This article does not contain any additional data.
Authors' contributions. Q.W., Y.H. and G.Z. conceptualized; Y.H. contributed to methodology and formal analysis; Q.W. contributed for software; J.W. and Z.W. validated the study; J.W. investigated; R.G. contributed to resources; Q.W. and Y.H. were involved in data curation; Q.W. contributed to original draft preparation; R.G., G.Z., Z.W. and J.W. reviewed and edited; J.W. contributed to visualization; H.Z. supervised the work and contributed to project administration; H.Z. and Y.Y. contributed to funding acquisition.
Funding. This research was funded by the National Natural Science Foundation of China (NSFC no. U1709201) and Shandong Province Key Research and Development Plan (grant no. 2019GGX102047).
Competing interests. The authors declare no conflict of interest.
Acknowledgements. The authors acknowledge the researchers in Shandong Haihua Group for their valuable discussions during this study.

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
