## [Reviewer comments · Royal Society Open Science]

Review History

RSOS-191138.R0 (Original submission)

Review form: Reviewer 1

Is the manuscript scientifically sound in its present form?

Yes

Are the interpretations and conclusions justified by the results?

Yes

Is the language acceptable?

No

Do you have any ethical concerns with this paper?

No

Have you any concerns about statistical analyses in this paper?

No

Recommendation?

Major revision is needed (please make suggestions in comments)

Comments to the Author(s)

The revision of the manuscript is required before acceptance and my comments are below

1. novelty be discussed
2. Have you checked selectivity of metal ions? How to find out quantitative analysis in the presence of metal ions.
3. How you explain mechanism without any supporting data?.
4. Did you the reversibility of sensor?
5. Following references must be included as literature is poor:
 Combinatorial Chemistry & High Throughput Screening, 14(4)(2011) 284-302.
 Journal of Molecular Liquids, Volume 177, January 2013, Pages 114-118
 Critical Reviews in Analytical Chemistry, 41(2011)282-313..
 Analyst, 120(1995)495-498.
 Analytical Proceedings including Analytical Communications 32, 21-23 (1995).
 Int. J.Electrochem.Sci, 10 (2015) 303-316.
 Sens. Actuators B195 (2014) 98-108.
 Anal. Chem., 68(1996)1272-1275.
 Journal of Molecular Liquids 174, 11-16,2011
 Sensors and Actuators B: Chemical 195, 98-108,2014
 J. Mol. Liq., 195 (2014)65-68.
 J. Mol. Liquids 173(2012)153-163.
 Journal of Molecular Liquids 215, 671-679; 2016
 RSC Adv., 5 (2015) 18438 - 18450.
 Water Research Volume 48, Issue 1, 1 January 2014, Pages 210-217
 Electrochimica Acta,120 (2014) 204-211.
 Sensors and Actuators, B: Chemical Volume 207, Issue Part A, February 2015, Pages 216-223
 Talanta, 144(2015) 80-89
 Ind. Eng. Chem. Res., 54 (14)(2015) 3634-3639.
 analytica Chimica Acta Volume 389, Issue 1-3, 14 May 1999, Pages 205-212
 Analytical Proceedings including Analytical Communications 32(1995)99-101.
 Analytical Proceedings including Analytical Communications 32(1995)263-265.
 Analyst, 120(1995)495-498.
 Anal. Chem., 68(1996)1272-1275.
 Analytical Proceedings including Analytical Communications 32, 21-23 (1995).
 , Cadmium (II) - selective sensors based on dibenzo-24-crown-8 in PVC matrix, Anal. Chim. Acta,
 389, 205-212 (1999).

Review form: Reviewer 2

Is the manuscript scientifically sound in its present form?

Yes

Are the interpretations and conclusions justified by the results?

Yes

Is the language acceptable?

No

Do you have any ethical concerns with this paper?

No

Have you any concerns about statistical analyses in this paper?

No

Recommendation?

Accept with minor revision (please list in comments)

Comments to the Author(s)

To begin with, the manuscript needs to be edited by a native English speaker, and Chinese characters removed from figures.

I have followed the work of the group for years and I have no problem with the quality of the data. But, the presentation is another matter. For instance, the title may be confusing as it seems to suggest that the sensor is applied to extract Br but in fact the sensor is used to monitor the Br extraction system. Other minor problems:

1. DCS should be spelled out when first used;
2. ISE denotes ion selective electrode, not just ion electrode;

Decision letter (RSOS-191138.R0)

12-Aug-2019

Dear Dr Wang:

Title: Application of Multi-Electrochemical Sensor in Extracting Bromine from Seawater
Manuscript ID: RSOS-191138

The editor assigned to your manuscript has now received comments from reviewers. We would like you to revise your paper in accordance with the referee and Subject Editor suggestions which can be found below (not including confidential reports to the Editor). Please note this decision does not guarantee eventual acceptance.

Please submit your revised paper before 04-Sep-2019. Please note that the revision deadline will expire at 00.00am on this date. If we do not hear from you within this time then it will be assumed that the paper has been withdrawn. In exceptional circumstances, extensions may be possible if agreed with the Editorial Office in advance. We do not allow multiple rounds of revision so we urge you to make every effort to fully address all of the comments at this stage. If deemed necessary by the Editors, your manuscript will be sent back to one or more of the original reviewers for assessment. If the original reviewers are not available we may invite new reviewers.

RSC Associate Editor:
Comments to the Author:
(There are no comments.)

RSC Subject Editor:
Comments to the Author:
(There are no comments.)

Reviewers' Comments to Author:
Reviewer: 1

Comments to the Author(s)

The revision of the manuscript is required before acceptance and my comments are below

1. novelty be discussed
2. Have you checked selectivity of metal ions? How to find out quantitative analysis in the presence of metal ions.
3. How you explain mechanism without any supporting data?.
4. Did you the reversibility of sensor?
5. Following references must be included as literature is poor:
Combinatorial Chemistry & High Throughput Screening, 14(4)(2011) 284-302.
Journal of Molecular Liquids, Volume 177, January 2013, Pages 114-118
Critical Reviews in Analytical Chemistry, 41(2011)282-313..
Analyst, 120(1995)495-498.
Analytical Proceedings including Analytical Communications 32, 21-23 (1995).
Int. J.Electrochem.Sci, 10 (2015) 303-316.

Sens. Actuators B195 (2014) 98-108.
 Anal. Chem., 68(1996)1272-1275.
 Journal of Molecular Liquids 174, 11-16,2011
 Sensors and Actuators B: Chemical 195, 98-108,2014
 J. Mol. Liq., 195 (2014)65-68.
 J. Mol. Liquids 173(2012)153-163.
 Journal of Molecular Liquids 215, 671-679; 2016
 RSC Adv., 5 (2015) 18438 - 18450.
 Water Research Volume 48, Issue 1, 1 January 2014, Pages 210-217
 Electrochimica Acta,120 (2014) 204-211.
 Sensors and Actuators, B: Chemical Volume 207, Issue Part A, February 2015, Pages 216-223
 Talanta, 144(2015) 80-89
 Ind. Eng. Chem. Res., 54 (14)(2015) 3634-3639.
 analytica Chimica Acta Volume 389, Issue 1-3, 14 May 1999, Pages 205-212
 Analytical Proceedings including Analytical Communications 32(1995)99-101.
 Analytical Proceedings including Analytical Communications 32(1995)263-265.
 Analyst, 120(1995)495-498.
 Anal. Chem., 68(1996)1272-1275.
 Analytical Proceedings including Analytical Communications 32, 21-23 (1995).
 , Cadmium (II) - selective sensors based on dibenzo-24-crown-8 in PVC matrix, Anal. Chim. Acta, 389, 205-212 (1999).

Reviewer: 2

Comments to the Author(s)

To begin with, the manuscript needs to be edited by a native English speaker, and Chinese characters removed from figures.

I have followed the work of the group for years and I have no problem with the quality of the data. But, the presentation is another matter. For instance, the title may be confusing as it seems to suggest that the sensor is applied to extract Br but in fact the sensor is used to monitor the Br extraction system. Other minor problems:

1. DCS should be spelled out when first used;
2. ISE denotes ion selective electrode, not just ion electrode

Author's Response to Decision Letter for (RSOS-191138.R0)

See Appendix A.

Decision letter (RSOS-191138.R1)

16-Sep-2019

Dear Dr Wang:

Title: Monitor Application of Multi-Electrochemical Sensor in Extracting Bromine from Seawater
 Manuscript ID: RSOS-191138.R1

It is a pleasure to accept your manuscript in its current form for publication in Royal Society Open Science. The chemistry content of Royal Society Open Science is published in collaboration with the Royal Society of Chemistry.

RSC Associate Editor
Comments to the Author:
(There are no comments.)

Reviewer(s)' Comments to Author:

Appendix A

Authors' Reply to Reviewers' Comments on “Monitor Application of Multi-Electrochemical Sensor in Extracting Bromine from Seawater”

*Qiujin Wang, Jianbo Wu, Guochen Zhao, Yuanfeng Huang, Zhen Wang, Hao Zheng ,
Yifan Zhou, Ying Ye, Reza Ghomashchi*

Thanks for the comments from reviewers. We are pleased to revise the manuscript as commented which help us to improve our work effectively. The response to each comment is listed below. Please find them in red color. Thanks again.

Reviewer: 1

The revision of the manuscript is required before acceptance and my comments are below

Comment 1: novelty be discussed

Response 1: Thanks. Nowadays, more and more attention has been paid globally to chemical safety production and green production. In this study, the multi-parameter electrochemical sensor was successfully developed, with integrated all-solid-state pH electrode, Eh electrode and bromine ion selective electrode (Br-ISE). The pH electrode was used to control the addition amount of H₂SO₄ during the acidification of the brine, The Eh electrode was used to control the addition amount of Cl₂ during the oxidation stage and the addition amount of SO₂ during the absorption stage, and the Br-ISE was used to monitor the Br⁻ concentration change in the raw brine. The purpose of developing the sensor is to stably realize on-line monitoring of parameters and precisely control the addition of reactants in real producing process, which is expected to eliminate the errors of manual frequent titration test, decrease production risk and save human costs.

Comment 2: Have you checked selectivity of metal ions? How to find out quantitative analysis in the presence of metal ions.

Response 2: Thanks. The working mechanism of pH electrode is as follows, which means

metal ions could hardly disturb the electrode reaction in solution. There are mainly two equilibria to describe the H⁺ response of the IrO₂ electrode [1].

For the Eh electrode, the redox potential of the brine and waste liquor itself is measured, and the metal ion selectivity test of the Eh electrode is not necessary.

For all-solid bromine ion selective electrodes, as we investigated, the most serious interference is Cl⁻, not Mg²⁺, K⁺, Ga²⁺ metal cations in brine. In our previous publication, we determined the selectivity of bromine ion electrodes by fixed interference ion method (FIM). [2]

Please refer the details below.

Selectivity is one of the most important performance indicators of a sensor, and it usually determines whether the test is reliable. As the sensitive films, not only respond to the Br⁻, but have different degrees of response to other coexisting ion, therefore, the response of the film is only relatively selective, not absolutely specific. In this paper, the FIM (Fixed Interference Method) recommended by the International Union of Theoretical and Applied Chemistry (IUPAC) is used to determine the selectivity of interfering ions of bromide electrodes.

$$K_{i,j}^{pot} = \frac{a_i}{a_j^{n_i/n_j}}$$

Where i is the ion to be measured, j is the interference ion, and a_i, a_j are the activity of the ion to be measured and the interference ion, respectively. n_i, n_j are the number of charges of the i and j respectively.

A mixed solution of a target analyte ion (Br⁻) and constant concentration of interfering ions are prepared. The main interfering ions are selected including (Cl⁻, I⁻, F⁻, NO₃⁻, NO₂⁻, SO₄²⁻, SO₃²⁻, SCN⁻) are fixed at 10⁻² M. In Table 1, the selectivity coefficients of the electrodes are listed to compare with previous reports. As can be seen table.1, the selectivity coefficients of Cl⁻ is lowest of all, for others also keep in a low level, which means the as-prepared Br-ISE is generally superior to the best bromide ion-selective electrodes reported in literature.

Table 1. Selectivity coefficient of various interfering anions.

K Ref..NO	Cl ⁻	I ⁻	F ⁻	NO ₃ ⁻	NO ₂ ⁻	SO ₄ ²⁻	SO ₃ ²⁻	SCN ⁻
3	8×10 ⁻²	3.98	—	—	—	—	—	—
4	1.2×10 ⁻²	1.8×10 ⁻³	2×10 ⁻⁵	2×10 ⁻⁵	1×10 ⁻⁵	1.5×10 ⁻⁵	2×10 ⁻⁵	1×10 ⁻³
5	8.6×10 ⁻²	6.4×10 ⁻²	4.2×10 ⁻⁵	8.4×10 ⁻⁴	1.2×10 ⁻⁵	3.2×10 ⁻⁴	—	1×10 ⁻³
6	1.4×10 ⁻³	5.5×10 ⁻³	7.5×10 ⁻⁴	8.5×10 ⁻⁴	8×10 ⁻⁴	3×10 ⁻⁴	3.1×10 ⁻⁴	7.5×10 ⁻³
7	9.0×10 ⁻⁴	1×10 ⁻³	2.6×10 ⁻⁵	8×10 ⁻³	4×10 ⁻³	2×10 ⁻⁴	—	3×10 ⁻³
8	7.9×10 ⁻²	2×10 ⁻²	—	8.9×10 ⁻²	6×10 ⁻³	3×10 ⁻³	—	6.9×10 ⁻²
This work	4.7×10 ⁻⁴	3.7×10 ⁻³	6.8×10 ⁻⁵	3.4×10 ⁻⁴	5.2×10 ⁻⁴	4.3×10 ⁻⁵	6.5×10 ⁻⁵	7.3×10 ⁻³

Comment 3: How you explain mechanism without any supporting data?

Response 3: Thanks for your comment. In fact, the explanation of mechanism was based on our previous published work. In this manuscript, we did not detailly study the mechanism since the mechanism was well established and suitable for the electrodes in the current study. The purpose to publish this study is to provide a strategy in sensor controlled automatic production process for but are not limited to the bromine factories.

Comment 4: Did you the reversibility of sensor?

Response 4: Thanks. Because the emphasis of this paper is on the application of sensors in real production of bromine from seawater, the Eh electrode of pH electrode has shown good performance in the previous application, and the repeatability of the bromine ion electrode developed for the extraction of bromine from seawater has been discussed.[9][10]

The Br-ISE is calibrated 9 times at different times, its Nernst slope remains basically unchanged, which also shows good repeatability.

Comment 5: Following references must be included as literature is poor:

Response 5: We have supplemented the relevant references in the manuscript.

Reviewer: 2

To begin with, the manuscript needs to be edited by a native English speaker, and Chinese characters removed from figures.

I have followed the work of the group for years and I have no problem with the quality of the data. But, the presentation is another matter. For instance, the title may be confusing as it seems to suggest that the sensor is applied to extract Br but in fact the sensor is used to monitor the Br extraction system. Other minor problems:

Thanks for your suggestion. We carefully edited the manuscript again. The grammar errors have been revised and the presentation has been improved in order to enhance its readability. As for the title, the sensor is indeed used to monitor the Br extraction system. Therefore we changed the title of the manuscript as “Monitor Application of Multi-Electrochemical Sensor in Extracting Bromine from Seawater”.

Comment 1: DCS should be spelled out when first used;

Response 1: DCS means Distributed Control System. We spelled out in the manuscript when

first used as you suggested. Thank you.

Comment 2: ISE denotes ion selective electrode, not just ion electrode;

Response 2: We just realized this error in abstract. The phrase “bromide ion electrode (Br-ISE)” has been revised as “bromide ion selective electrode (Br-ISE)”. Many thanks.

References:

- [1] Bestaoui B, Prouzet E, Deniard P, et al. Structural and analytical characterization of an iridium oxide thin layer[J]. *Thin Solid Films*, 1993, 235: 35–42
- [2] Malik, R.; Zhang, L.; McConnell, C. Three-dimensional, free-standing polyaniline/carbon nanotube composite-based electrode for high-performance supercapacitors. *Carbon* 2017, 116, 579–590.
- [3] Si, W.; Lei, W.; Han, Z.; Hao, Q.; Zhang, Y.; Xia, M. Selective sensing of catechol and hydroquinone based on poly(3,4-ethylenedioxythiophene)/nitrogen-doped graphene composites. *Sens. Actuators B Chem.* 2014, 199, 154–160.
- [4] Wang, Z.L.; Xu, D.; Wang, H.G.; Wu, Z.; Zhang, X.B. In situ fabrication of porous graphene electrodes for high-performance energy storage. *ACS Nano* 2013, 7, 24
- [5] Peng, S.; Yan, X.; Zhang, D.; Wu, X.; Luo, Y.; He, G. A H₃PO₄ preswelling strategy to enhance the proton conductivity of a H₂SO₄ -doped polybenzimidazole membrane for vanadium flow batteries. *RSC Adv.* 2016, 6, 23479–23488.
- [6] Rius-Ruiz, F.X.; Kisiel, A.; Michalska, A.; Maksymiuk, K.; Riu, J.; Rius, F.X. Solid-state reference electrodes based on carbon nanotubes and polyacrylate membranes. *Anal. Bioanal. Chem.* 2011, 399, 3613–3622.
- [7] Ruiz, D.; del Rosal, B.; Acebrón, M.; Palencia, C.; Sun, C.; Cabanillas-González, J.; López-Haro, M.; Hungría, A.B.; Jaque, D.; Juarez, B.H. Ag/Ag₂S nanocrystals for high sensitivity near-infrared luminescence nanothermometry. *Adv. Funct. Mater.* 2017, 27, 160462.
- [8] Qiuji Wang, Yifan Zhou, Jixue Zhou, *Coatings*, 2019, 9, 325-339.
- [9] Wu, Rongrong, Tao, Chunhui, Chen, Xuegang, Ye, Ying, Yue, Xihe, Huang, Yuqiang, A Multi-parameter Chemical Sensor and its Application in Hydrothermal Exploration of the Southwest Indian Ridge, *Marine Georesources & Geotechnology*,
- [10] Zhang X, Ye Y, Kan Y, et al. A new electroplated Ir/Ir(OH)_x pH electrode and its application in the coastal areas of Newport Harbor, California[J]. *Acta Oceanol. Sin.*, 2017, 36, 99–104.